# Ten-Year Trends in Psychotropic Prescribing and Polypharmacy in Australian General Practice Patients with and without Dementia

**DOI:** 10.3390/jcm12103389

**Published:** 2023-05-10

**Authors:** Woldesellassie M. Bezabhe, Jan Radford, Mohammed S. Salahudeen, Ivan Bindoff, Tristan Ling, Peter Gee, Barbara C. Wimmer, Gregory M. Peterson

**Affiliations:** 1School of Pharmacy and Pharmacology, University of Tasmania, Private Bag 26, Hobart, TAS 7001, Australia; 2Launceston Clinical School, Tasmanian School of Medicine, University of Tasmania, 41 Frankland St, Launceston, TAS 7250, Australia

**Keywords:** psychotropic drugs, prescribing, polypharmacy, dementia, primary care, behavioural and psychological symptoms of dementia

## Abstract

Objective: Little research has evaluated trends in psychotropic prescribing and polypharmacy in primary care patients, especially those with dementia. We sought to examine this in Australia from 2011 to 2020 using the primary care dataset, MedicineInsight. Methods: Ten consecutive serial cross-sectional analyses were performed to evaluate the proportion of patients aged 65 years or more, with a recorded diagnosis of dementia, who were prescribed psychotropic medications within the first six months of each year from 2011 to 2020. This proportion was compared with propensity score-matched control patients without dementia. Results: Before matching, 24,701 patients (59.2% females) with, and 72,105 patients (59.2% females) without, a recorded diagnosis of dementia were included. In 2011, 42% (95% confidence interval [CI] 40.5–43.5%) of patients in the dementia group had at least one recorded prescription of a psychotropic medication, which declined to 34.2% (95% CI 33.3–35.1%; *p* for trend < 0.001) by 2020. However, it remained unchanged for matched controls (36% [95% CI 34.6–37.5%] in 2011 and 36.7% [95% CI 35.7–37.6%] in 2020). The greatest decline in the dementia groups by medication class was for antipsychotics (from 15.9% [95% CI 14.8–17.0%] to 8.8% [95% CI 8.2–9.4%]; *p* for trend < 0.001). During this period, the prevalence of psychotropic polypharmacy (use of two or more individual psychotropics) also decreased from 21.7% (95% CI 20.5–22.9%) to 18.1% (95% CI 17.4–18.9%) in the dementia groups, and slightly increased from 15.2% (95% CI 14.1–16.3%) to 16.6% (95% CI 15.9–17.3%) in the matched controls. Conclusions: The decline in psychotropic prescribing, particularly antipsychotics, in Australian primary care patients with dementia is encouraging. However, psychotropic polypharmacy still occurred in almost one in five patients with dementia at the end of the study period. Programs focused on encouraging further reductions in the use of multiple psychotropic drugs in patients with dementia are recommended, particularly in rural and remote regions.

## 1. Introduction

Over 55 million people live with dementia globally [1]. In Australia, over 436,000 people live with dementia [2,3]; two-thirds live in the community [3]. One in 10 people over 65 years has a diagnosis of dementia, and dementia is the second leading cause of death [4]. Behavioural and psychological symptoms of dementia (BPSD) occur at any stage in dementia due to Alzheimer’s disease (AD) and early in frontotemporal dementia or dementia with Lewy bodies, and their management is a major challenge. These symptoms can include agitation, aggression, depression, anxiety, delusions, and hallucinations [5]. BPSD are associated with negative outcomes, such as decreased quality of life, hospitalisations, and increased healthcare costs, although they often subside spontaneously within 6 months [6].

Psychotropic drugs are a second-line option to psychosocial interventions for managing BPSD [7]. These drugs include antipsychotics, antidepressants, anxiolytics, and hypnotics; in addition, anticonvulsants and opioids also fall within the psychotropic grouping [8]. Evidence regarding the efficacy of psychotropic medicines in patients with BPSD is inconsistent. Antipsychotics are effective in only approximately one in five people with dementia for short-term management of significant agitation, aggression, and psychosis. These medications (especially antipsychotics or benzodiazepines) are associated with negative outcomes, such as a decline in cognition and increased risk of falls, stroke, and all-cause death [5,7]. Patients with dementia with Lewy bodies may experience severe adverse reactions when taking antipsychotics. Acetylcholinesterase inhibitors (rivastigmine or donepezil) are the drug of choice for agitation, aggression, or psychosis in these patients [7].

Previous studies on the use of psychotropic medications in Australia are limited to long-term (or residential aged) care facilities [9,10,11,12,13,14,15]. A study by Westbury et al. [12] reported nearly two-thirds (61%) of people in Australian residential aged care were taking psychotropic drugs regularly; of these, 25.6% were prescribed psychotropic polypharmacy (two or more psychotropic drugs), which is not recommended in older people [16,17]. That study used data from community pharmacies where patient diagnoses, including dementia, were not recorded; thus, it did not examine psychotropic prescribing specifically in patients with dementia.

In recent years, inappropriate use of psychotropics in the elderly, particularly in those with dementia, has been noted as a significant concern. This has given rise to collaborative efforts to address the issue, such as interdisciplinary interventions [13,18], recommendations from the Australian Royal Commission into Aged Care Quality and Safety [19], and updated guidelines [7,20]. While there are studies examining the impact of some of these efforts in residential aged care, there is a need for a nationally representative study examining psychotropic prescribing and polypharmacy in primary care patients with and without a recorded dementia diagnosis. Using Australian general practice data, we sought to examine the prevalence and temporal trends in the use of psychotropic drugs in patients with a recorded dementia diagnosis compared with those without dementia.

## 2. Methods

We used data from MedicineInsight, a dataset extracted and collated by NPS MedicineWise from electronic health records (EHRs) of its enrolled general practices across Australia. The EHRs are maintained by general practitioners (GPs) during the provision of usual clinical care. They contain sociodemographic information (e.g., sex, age, rurality, and socioeconomic status), diagnoses, prescriptions, clinical encounters, observations, and laboratory tests. Unstructured data in the EHRs’ ‘progress notes’, which may contain patient-specific information, were not extracted. The data of 423 general practices across Australia met the standard data quality criteria (described elsewhere) [21] and were included. The dataset represents the Australian population in terms of sex and age [21].

Specifically for this study, we obtained a record of all patients with a documented diagnosis of dementia (n = 47,158) and controls (n = 137,120) (selected by matching with dementia cases by sex, age, state, and general practice site) from MedicineInsight. Only patients who had at least three recorded encounters at the same general practice between 1 January 2011 and 31 December 2020 were included, for both patients with dementia (n = 24,701) and controls (n = 72,105). Patients whose date of diagnosis of dementia was not recorded or was after 31 December 2019 were excluded. We required patients with dementia to have at least one year of data after their recorded dementia diagnoses date. Details of patient inclusion are shown in Figure 1. For each year of analysis, we only included regular patients (having at least three recorded practice visits in the previous two years before 1 January of the census year) who were aged 65 years or more [22]. The age of each patient was estimated from their year of birth (using 1 July in the patient’s recorded year of birth).

We constructed a dementia group and matched control group for each year using a 1:1 pair propensity score, matching patients with dementia with those of controls. The matching covariates were sex, age, rurality [23], socioeconomic indexes for areas (SEIFA) [23], congestive heart failure, hypertension, stroke, vascular disease, anxiety, arthritis (flag indicating arthritis, rheumatoid arthritis or osteoarthritis), asthma, deep vein thrombosis, depression (or bipolar disorder), cancer, coronary heart disease, chronic liver disease, chronic obstructive pulmonary disease, atrial fibrillation, atrial flutter, substance abuse, epilepsy, osteoporosis, chronic pain (flag indicating lower back pain or chronic pain), and schizophrenia. The conditions included those having a compelling indication for the use of psychotropic medications so that any difference in psychotropic prescribing and polypharmacy between the groups would reflect use associated with dementia. MedicineInsight flagged patient conditions were used for matching; the coded and non-coded terms used to identify these conditions are found in the MedicineInsight Dictionary [24] and previous publications [25]. The Australian Bureau of Statistics (ABS) developed the SEIFA quintile index, which ranks areas in Australia from 1 (most disadvantaged) to 5 (most advantaged) [23]. The ABS also categorised rurality into five categories using the Accessibility/Remoteness Index of Australia (ARIA) score. These categories include major cities (ARIA 0–0.20), inner regional (0.21–2.40), outer regional (2.41–5.92), remote (5.93–10.53), and very remote (10.54–15) [26].

The propensity score 1:1 matching occurred in descending order without replacement. A calliper width of 0.01 on the logit of propensity score was used for matching [27,28]. An absolute standardised difference of ≥0.10 was considered a significant imbalance between the groups. As an indication, the total number of patients with dementia and matched controls for every three years of the study period are shown in Table 1.

Psychotropic polypharmacy was defined as at least two recorded prescriptions of unique medications within the first six months of the year from the following six classes of medications: antipsychotics (ATC code N05A), anxiolytics (N05B), hypnotics (N05C), antidepressants (N06A), opioids (N02A), or antiepileptics (N03A) [16]. These classes of medications were selected based on the updated Beers Criteria from 2019 [29].

Ten consecutive serial cross-sectional analyses were performed to calculate the proportion (with 95% confidence intervals (CI)) of patients with dementia aged 65 years or more who were prescribed psychotropic polypharmacy from 1 January 2011 to 31 December 2020. This proportion was compared with propensity score-matched control patients without dementia. Temporal trends were shown in graphs, and we used a Cochran-Armitage test for trends to ascertain statistical significance [30].

Sensitivity Analysis

A sensitivity analysis was conducted to examine psychotropic polypharmacy for the study period for both dementia and control groups without matching.

Statistical Analysis

SAS software (SAS version 9.4, SAS Institute Inc., Cary, NC, USA) was used for all data analyses, and a two-sided *p*-value < 0.05 was considered statistically significant.

## 3. Results

Before matching, 24,701 patients with recorded dementia were included; of these, 14,619 (59.2%, 95% CI 58.6–59.8%) were females. We also included 72,105 (42,701, 59.2%, 95% CI 58.9–59.6% female) patients without a recorded diagnosis of dementia as the control group (Figure 1). The total number of patients aged 65 years or more included in the dementia groups each year ranged from 4400 in 2011 to 9825 in 2020 (Table 1). The matched patient characteristics are displayed every three years (2011, 2014, 2017, and 2020). As we used 1:1 pair matching, the total number of patients in the matched control for each index year was the same as the dementia group.

The 20 most commonly prescribed individual psychotropic medications in patients with dementia in 2020 are shown in Table 2. Of these, 6 were antidepressants, 4 were antipsychotics, and 4 were opioids. The top five individual psychotropic medications were oxycodone (5.7%), mirtazapine (5.6%), buprenorphine (5%), escitalopram (3.8%), and risperidone (3.7%).

The key changes in psychotropic medication use over time are shown in Figure 2 and Appendix A. The prevalence of antipsychotic drug use in the dementia groups was at least double that of matched controls between 2011 and 2020. However, over this time, it declined in both groups, from 15.9% (95% CI 14.8–17.0%) to 8.8% (95% CI 8.2–9.4%; *p* for trend < 0.001) in the dementia group and 6.2% (95% CI 5.5–7.0%) to 4.5% (95% CI 4.1–4.9%; *p* for trend < 0.001) in the matched controls (Figure 2A). With sub-analysis based on age, antipsychotic prescribing in the dementia groups declined in all three age groups (65–74, 75–84, and ≥85 years) and from 19.8% (95% CI 16.1–24.0%) in 2011 to 12.8% (95% CI 10.9–14.9%) in 2020 for those 85 years or more. It remained highest in people aged 85 years or older (Figure 2B).

Antipsychotic prescribing was higher in females with dementia (16.7%) compared to males (14.4%) in 2011 and declined in both groups with a similar rate (approximately 9%) by 2020. In controls, females (8.3% in 2012 to 4.9% in 2020) maintained a higher rate of antipsychotic use than males (4.8% in 2011 to 3.3% in 2020) (Figure 2C).

The use of opioids was slightly higher in the dementia groups compared with the matched controls in all study years (Appendix A). Prescribing of hypnotics for all study periods and anxiolytics between 2016 and 2020 was higher in the matched controls than in the dementia groups (Appendix A).

In 2011, 42% (95% CI 40.5–43.5%) of patients in the dementia group had at least one recorded prescription of psychotropic medication, which declined to 34.2% (95% CI 33.3–35.1%; *p* for trend < 0.001) by 2020 (Figure 3). The mean (standard deviation) number of individual psychotropic medications in those with at least one psychotropic medication was 1.9 (1.7) in 2011 and 2.0 (1.8) in 2020 for the dementia group, and 1.7 (1.0) in 2011 and 1.8 (1.1) in 2020 for the matched controls. Between 2011 and 2020, the prevalence of psychotropic polypharmacy (having ≥2 recorded prescriptions from the six individual classes of psychotropic medications mentioned above) was higher in the dementia groups compared with the matched controls (Figure 3). However, it decreased from 21.7% (95% CI 20.5–22.9%) to 18.1% (95% CI 17.4–18.9%; *p* for trend < 0.001) in the dementia groups and slightly increased from 15.2% (95% CI 14.1–16.3%) to 16.6% (95% CI 15.9–17.3%; *p* for trend < 0.001) in the matched controls. Prescribing three or more psychotropic medications dropped slightly in the dementia groups from 9.6% (95% CI 8.7–10.5%) in 2011 to 8.5% (95% CI 8.0–9.1%) in 2020, while it increased in matched controls from 5.7% (95% CI 5.0–6.4%) in 2011 to 7.0% (95% CI 6.5–7.6%) in 2020 (Figure 3). The trend in psychotropic polypharmacy calculated using unmatched dementia and control groups (sensitivity analysis) was similar to the one obtained from matched groups (Appendix A).

Over the 10 years, psychotropic polypharmacy was higher in females for both dementia and control groups (Figure 4A). It declined from 23.1% (95% CI 21.5–24.7%) in 2011 to 18.4% (95% CI 17.4–19.4%; *p* for trend < 0.001) in 2020 in female patients with dementia, while it slightly increased from 17.2% (95% CI 15.9–18.7%) to 19.6% (95% CI 18.6–20.7%) in the female matched controls. Although psychotropic polypharmacy was lower in males, the gap between male dementia and male matched controls was large. In 2011, psychotropic polypharmacy occurred in 19.3% (95% CI 17.4–21.3%) of the male dementia group compared with 11.5% (95% CI 10.0–13.2%) of the male matched control group. The gap persisted at the end of the study period, when 17.8% (95% CI 16.6–19.0%) of the male dementia patients had psychotropic polypharmacy, compared with 12.4% (95% CI 11.4–13.4%) of the male matched control group.

Psychotropic polypharmacy was high and remained unchanged in patients with dementia aged ≥85 years: 24.6% (95% CI, 21.0–29.0%) in 2011 and 25.1% (95% CI 22.6–27.7%) in 2020. Further sub-analysis by psychotropic class is shown in Appendix A. The decline in antipsychotic prescribing (from 19.8% [95% CI, 16.2–23.9%] in 2011 to 12.8% [95% CI, 10.9–14.9%] in 2020) in the dementia groups aged 85 years or older was compensated by slight increases in prescribing of opioids (from 11.8% [95% CI, 8.8–15.2%] in 2011 to 14.2% [95% CI, 12.2–16.3%] in 2020), antiepileptics (from 8.1% [95% CI, 5.7–11.0%] in 2011 to 11.2% (95% CI, 9.4–13.1%] in 2020), and antidepressants (from 27.4% [95% CI, 23.3–31.9%] in 2011 to 29.5% [95% CI, 26.9–32.2%] in 2020). In the matched control groups aged 85 years or older, psychotropic polypharmacy increased from 12.9% (95% CI 10.0–16.6%) in 2011 to 18.0% (95% CI 15.8–20.5%) in 2020. It significantly declined in those with dementia aged 65–74 years from 20.3% (95% CI 18.6–22.1%) in 2011 to 15.0% (95% CI, 14.0–16.0%; *p* for trend < 0.001) in 2020 (Figure 4B).

## 4. Discussion

Although not extensively examined in elderly community-based Australian patients with dementia, the problem of psychotropic prescribing for patients with dementia has been extensively reported in international studies [17,31,32]. van der Speak et al. developed an index that measured the appropriateness of psychotropic drug use in patients with dementia [33] that was implemented in a study of 380 patients for whom intervention, based upon the use of that index, was shown to produce favourable results [34]. The three main findings are highlighted below. Firstly, over the study period, psychotropic prescribing in patients with dementia declined. Secondly, the trends in prescribing varied by class of psychotropics. The highest decline was for antipsychotic medications, followed by hypnotics and anxiolytics. The changes in individual classes of psychotropic medications are discussed below. Thirdly, psychotropic polypharmacy was more likely in patients with dementia who were female, of older age (≥85 years) and living in regional and remote Australia.

Antipsychotic use in patients with dementia nearly halved during the study period (from a prevalence of 15.8% in 2011 to 8.8% in 2020). This is in line with a recent report by the Australian Commission on Safety and Quality in Health Care, showing an 11% reduction in antipsychotic dispensing for all people aged 65 years or more between 2016–2017 and 2020–2021 [35].

A declining trend in antipsychotic prescribing has also been observed internationally [28] and is attributed to the release of guidelines, safety warnings, and tightened prescribing restrictions [36]. For instance, in Australia, the Royal Commission into Aged Care Quality and Safety was established in 2018 and subsequently released statements with recommendations to restrict the potentially inappropriate use of psychotropic medicines [8].

Over the study period, overall use of anxiolytics and hypnotics declined and was lower in patients with dementia than those without dementia. This is in line with a recent analysis of MedicineInsight for the use of benzodiazepines and non-benzodiazepine benzodiazepine receptor agonist hypnotics (Z-drugs) in all patients aged 18 years and over [37], and international studies elsewhere [38,39]. Several reasons might be associated with this decreasing trend in prescribing. One might be related to warnings and the release of safety guidelines, as pointed out above, against prescribing these drugs to patients with dementia [7,8].

Psychotropic polypharmacy in the dementia groups declined from 21.7% in 2011 to 18.1% in 2020. It was lower than the 25.6% [9] reported by a previous Australian study and the 33% (95%CI 28–39%) [17] reported by an international meta-analysis involving 25 studies from 12 countries (defining psychotropic polypharmacy as the use of two or more individual psychotropics) in dementia patients in residential aged care.

Females, older people, and those living in more disadvantaged areas (regional, outer regional, remote, and very remote Australia) were more likely to be prescribed psychotropic polypharmacy. Previous studies reported higher psychotropic polypharmacy in females and related this with their more frequent BPSD symptoms [40]. However, non-pharmacological interventions are preferred over psychotropic medications, which have minimal efficacy and are associated with an increased risk of fall-related injury, sedation, hospitalisation, stroke, and death in older people with dementia [41]. Moreover, a recent cross-national European study that compared BPSD by gender in aged care residents with cognitive impairment found that males were more likely to exhibit these symptoms (wandering, verbal and physical abuse, socially inappropriate behaviour, and sexually uninhibited behaviour) than females, in whom only depression was more likely [31]. We acknowledge psychotropic polypharmacy is not always inappropriate in patients with dementia, given that conditions, including depression and pain, have often been under-managed in these patients [42].

Overall, the use of antidepressants, antiepileptics, and opioids was higher in patients with dementia than in those without. Their increased prescribing in older people, aged 85 years or more, prevented psychotropic polypharmacy from declining by compensating for the reduced antipsychotic prescribing. A previous Australian study [12] in residential aged care found that 16% of residents were prescribed antidepressants in 2014–2015 (compared with 18.7% in 2020 in this study). The antidepressant doses prescribed for most residents in that study were lower than recommended to treat depression and were taken at night. This suggests that antidepressants were often taken for their sedating effect. However, antidepressants are associated with adverse outcomes. For instance, mirtazapine, the most commonly prescribed antidepressant in our study sample, increased the risk of falls and stroke in older people [43] and had no benefit in treating agitated behaviours in people with dementia [44].

The increase in opioid prescribing that coincided with the decline in antipsychotic prescribing may suggest that opioids, to some extent, replaced antipsychotics for managing BPSD. A recent Danish study found a similar trend of increasing opioid prescribing and decreasing antipsychotic prescribing in people with dementia aged 65 years and over [32]. Recognising and managing pain in patients with dementia has received substantial attention over the last ten years. This could also be a reason for the increased opioid use [32].

## 5. Strengths and Limitations

Using a large Australian-wide dataset, this study is the first to describe 10-year trends in psychotropic prescribing and polypharmacy in primary care patients with dementia. The matching of dementia and control patients was comprehensive, including conditions that compel an indication for psychotropic therapy. Therefore, the difference in the prevalence of psychotropic prescribing between patients with dementia and controls should largely reflect the use for managing BPSD. However, this study had several limitations. The severity of dementia was not recorded, and psychotropic prescribing was therefore not examined based on dementia severity. Acetylcholinesterase inhibitors use associated with lowered risk of antipsychotic and anxiolytic initiation [45]. This study did not examine whether the use of acetylcholinesterase inhibitors influenced psychotropic prescribing.

We used recorded prescriptions and cannot confirm whether these medications were dispensed or taken. Prescriptions written by non-GP specialists (e.g., psychiatrists or geriatricians) were not recorded in the MedicineInsight dataset. However, GPs in Australia usually continue those prescriptions, so the trends in psychotropic prescribing and polypharmacy presented in this study are likely to be accurate. It is a notable strength of the present study that the controls and patients were matched by using a most extensive list of co-morbidities; and a particular strength is the finding that in those age ≥85, those with dementia had a ~25% rate of psychotropic polypharmacy as compared with ~15% in the controls (Figure 4B).

## Figures and Tables

**Figure 1 jcm-12-03389-f001:**
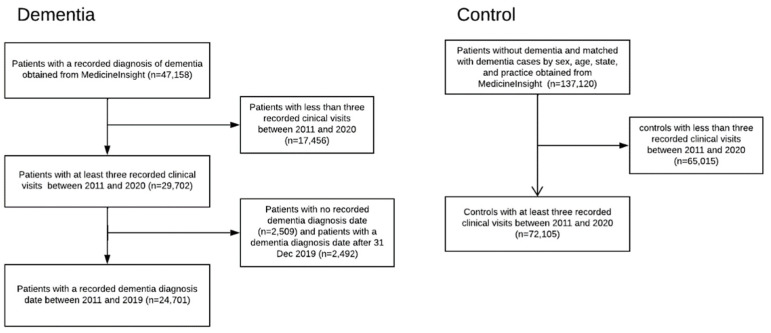
Patient selection flow chart.

**Figure 2 jcm-12-03389-f002:**
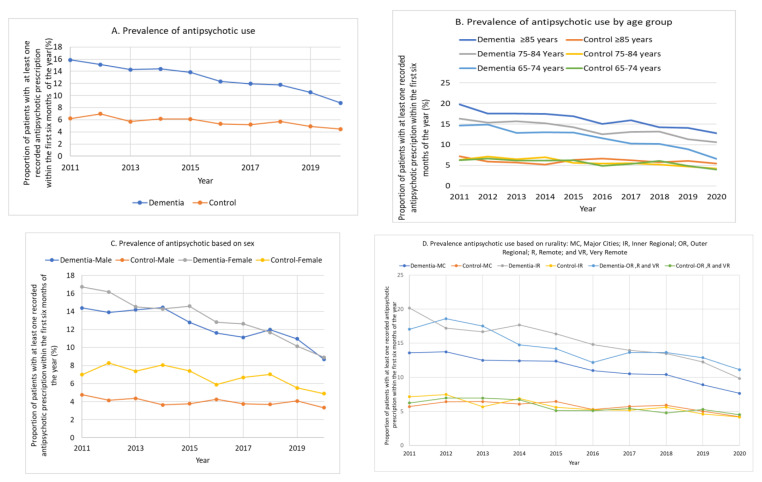
Trends in antipsychotic prescribing in the dementia groups compared with their matched controls (**A**) based on age (**B**), sex (**C**), and rurality (**D**).

**Figure 3 jcm-12-03389-f003:**
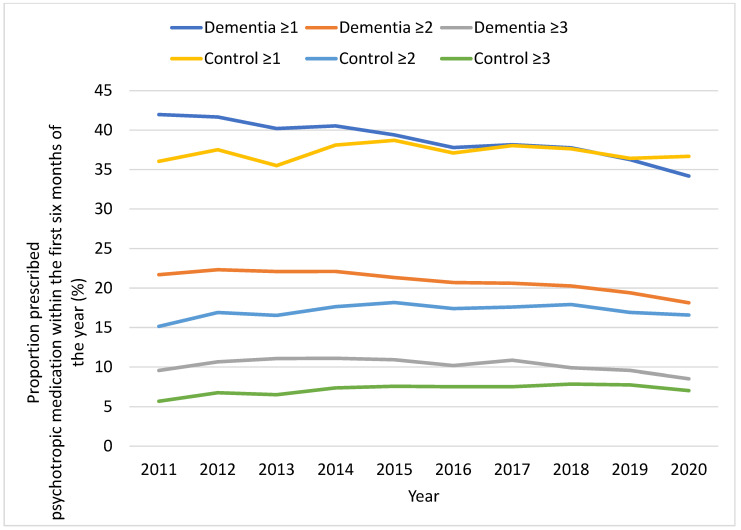
Prevalence of psychotropic prescribing in primary care patients with dementia and their matched controls, based on the number of psychotropics prescribed within six months: at least 1 (≥1), at least 2 (≥2), and at least 3 (≥3).

**Figure 4 jcm-12-03389-f004:**
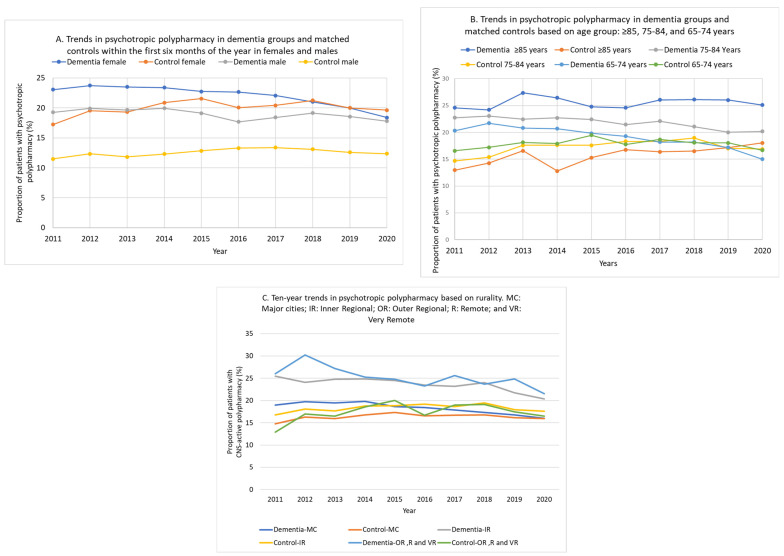
Trends in psychotropic polypharmacy in dementia groups and their matched controls based on sex (**A**), age (**B**), and rurality (**C**). With sub-analysis based on rurality (regrouped into three classes: major cities, inner regional, and outer regional, remote, and very remote), during the study period, psychotropic polypharmacy was higher in patients with dementia living in inner regional or outer regional, remote, and very remote Australia, than those living in major cities. The highest prevalence of psychotropic polypharmacy in the dementia group living in outer regional, remote, and very remote Australia was 30.3% (95% CI 26.7–34.0%) in 2012, declining to 21.5% (95% CI 19.4–23.8%; *p* for trend < 0.001) by 2020. The prevalence of psychotropic polypharmacy in dementia patients living in inner regional Australia peaked at 25.5% (95% CI 23.1–28.0%) in 2011 and declined to 20.4% (95% CI 19.0–21.9%; *p* for trend < 0.001) in 2020 (Figure 4C).

**Table 1 jcm-12-03389-t001:** Characteristics of patients with dementia and their matched controls every three years from 2011–2020.

Year	2011		2014		2017		2020	
	Dementia	Control	Dementia	Control	Dementia	Control	Dementia	Control
Sample (n)	4400	4400	6722	6722	8367	8367	9825	9825
Age (mean (SD))	83.7 (6.7)	83.9 (6.9)	84.2 (7.1)	84.2 (7.2)	84.0 (7.4)	84.0 (7.2)	84.0 (7.5)	84.1 (7.2)
≥85	2132 (48.5)	2168 (49.3)	3426 (51.0)	3406 (50.7)	4235 (50.6)	4221 (50.5)	4969 (50.6)	4978 (50.7)
75–85	1834 (41.7)	1804 (41.0)	2592 (38.6)	2683 (39.9)	3153 (37.7)	3248 (38.8)	3684 (37.5)	3777 (38.4)
65–74	434 (9.9)	428 (9.7)	704 (10.5)	633 (9.4)	979 (11.7)	898 (10.7)	1172 (11.9)	1070 (10.9)
Sex-female (%)	2810 (63.9)	2810 (63.9)	4175 (62.1)	4175 (62.1)	4995 (59.7)	4995 (59.7)	5728 (58.3)	5728 (58.3)
**Rurality (%)**								
Major cities	2615 (59.4)	2624 (59.6)	3729 (55.5)	3759 (55.9)	4614 (55.2)	4511 (53.9)	5411 (55.1)	5286 (53.8)
Inner regional	1257 (28.6)	1274 (29.0)	2092 (31.1)	2076 (30.9)	2587 (30.9)	2718 (32.5)	3022 (30.8)	3234 (32.9)
Outer regional	519 (11.8)	480 (10.9)	874 (13.0)	848 (12.6)	1092 (13.1)	1081 (12.9)	1276 (13.0)	1232 (12.5)
Remote and very remote	9 (0.2)	22 (0.5)	27 (0.4)	39 (0.6)	74 (0.9)	57 (0.7)	116 (1.2)	73 (0.7)
**SEIFA quintiles (%)**								
1	754 (17.1)	836 (19.0)	1245 (18.5)	1344 (20.0)	1827 (21.8)	1835 (21.9)	2033 (20.7)	2065 (21.0)
2	1083 (24.6)	1012 (23.0)	1778 (26.5)	1680 (25.0)	1985 (23.7)	1950 (23.3)	2274 (23.2)	2314 (23.6)
3	714 (16.2)	697 (15.8)	1150 (17.1)	1114 (16.6)	1437 (17.2)	1478 (17.7)	1997 (20.3)	1811 (18.4)
4	775 (17.6)	762 (17.3)	1044 (15.5)	1037 (15.4)	1275 (15.2)	1308 (15.6)	1503 (15.3)	1513 (15.4)
5	1074 (24.4)	1093 (24.8)	1505 (22.4)	1547 (23.0)	1843 (22.0)	1796 (21.5)	2018 (20.5)	2122 (21.6)
**Australian State**								
New South Wales	1689 (38.4)	1759 (40.0)	2621 (39.0)	2654 (39.5)	3152 (37.7)	3160 (37.8)	3883 (39.5)	3982 (40.5)
Victoria	1037 (23.6)	979 (22.3)	1475 (21.9)	1467 (21.8)	1821 (21.8)	1809 (21.6)	1990 (20.3)	2029 (20.7)
Queensland	586 (13.3)	547 (12.4)	811 (12.1)	811 (12.1)	1277 (15.3)	1195 (14.3)	1678 (17.1)	1513 (15.4)
Western Australia	331 (7.5)	337 (7.7)	503 (7.5)	529 (7.9)	580 (6.9)	651 (7.8)	718 (7.3)	653 (6.7)
South Australia	70 (1.6)	91 (2.1)	142 (2.1)	191 (2.8)	170 (2.0)	191 (2.3)	185 (1.9)	200 (2.0)
Tasmania	553 (6.3)	591 (6.7)	952 (14.2)	915 (13.6)	1009 (12.1)	1104 (13.2)	1053 (10.7)	1180 (10.7)
Northern Territory	127 (2.9)	93 (2.1)	211 (3.1)	146 (2.2)	343 (4.1)	245 (2.9)	297 (3.0)	265 (2.7)
Australian Capital Territory	7 (0.2)	3 (0.1)	7 (0.1)	9 (0.1)	15 (0.2)	12 (0.1)	21 (0.2)	3 (0.0)
**Comorbidities**								
Heart failure	848 (19.3)	870 (19.8)	1327 (19.7)	1343 (20.0)	1567 (18.7)	1581 (18.9)	1637 (16.7)	1623 (16.5)
Hypertension	2739 (62.3)	2743 (62.3)	4374 (65.1)	4408 (65.6)	5572 (66.6)	5573 (66.6)	6612 (67.3)	6616 (67.3)
Stroke	1238 (28.1)	1248 (28.4)	1833 (27.3)	1875 (27.9)	2152 (25.7)	2136 (25.5)	2293 (23.3)	2306 (23.5)
Peripheral vascular disease	1527 (34.7)	1562 (35.5)	2291 (34.1)	2283 (34.0)	2744 (32.8)	2731 (32.6)	3131 (31.9)	3095 (31.5)
Deep vein thrombosis	66 (1.5)	52 (1.2)	92 (1.4)	85 (1.3)	143 (1.7)	148 (1.8)	159 (1.6)	145 (1.5)
Atrial fibrillation	828 (18.8)	866 (19.7)	1366 (20.3)	1364 (20.3)	1785 (21.3)	1779 (21.3)	2113 (21.5)	2120 (21.6)
Atrial flutter	44 (1.0)	40 (0.9)	80 (1.2)	91 (1.4)	115 (1.4)	113 (1.4)	145 (1.48)	130 (1.3)
Anxiety	729 (16.6)	715 (16.3)	1390 (20.7)	1418 (21.1)	1945 (23.3)	1997 (23.9)	2506 (25.5)	2526 (25.7)
Depression	1733 (39.4)	1699 (38.6)	2858 (42.5)	2825 (42.0)	3577 (42.8)	3594 (43.0)	4148 (42.2)	4130 (42.0)
Schizophrenia	119 (2.7)	83 (1.9)	164 (2.4)	132 (2.0)	148 (1.8)	129 (1.5)	166 (1.70)	130 (1.3)
Epilepsy	176 (4.0)	156 (3.6)	235 (3.5)	229 (3.4)	265 (3.2)	249 (3.0)	281 (2.9)	270 (2.8)
Arthritis	2454 (55.8)	2457 (55.8)	3982 (59.2)	3988 (59.3)	5164 (61.7)	5212 (62.3)	6055 (61.6)	6018 (61.3)
Osteoporosis	1390 (31.6)	1388 (31.6)	2157 (32.1)	2102 (31.3)	2936 (35.1)	2941 (35.2)	3624 (36.9)	3617 (36.8)
Asthma	540 (12.3)	508 (11.6)	869 (12.9)	802 (11.9)	1165 (13.9)	1152 (13.8)	1411 (14.4)	1374 (14.0)
Chronic obstructive pulmonary disease	592 (13.4)	579 (13.2)	959 (14.3)	941 (14.0)	1292 (15.4)	1257 (15.0)	1507 (15.3)	1472 (15.0)
Cancer	1809 (50.3)	1788 (49.7)	3149 (46.9)	3073 (45.7)	4191 (50.1)	4131 (49.4)	4813 (49.0)	4785 (48.7)
Chronic liver disease	17 (0.4)	11 (0.3)	31 (0.5)	34 (0.5)	44 (0.5)	59 (0.7)	62 (0.6)	68 (0.7)
Substance abuse	110 (2.5)	97 (2.20)	192 (2.9)	214 (3.2)	295 (3.5)	293 (3.5)	371 (3.8)	357 (3.6)
Pain	1179 (26.8)	1162 (26.4)	2134 (31.8)	2141 (31.9)	3153 (37.7)	3165 (37.8)	3896 (39.7)	3872 (39.4)

SD, standard deviation; SEIFA, socioeconomic indexes for areas.

**Table 2 jcm-12-03389-t002:** Twenty most frequently prescribed psychotropic medications in 12,890 patients with dementia in 2020.

Rank	Medicine Name	Patients Prescribed n (%)	ATC Code
1	Oxycodone	737 (5.7)	N02AA05, N02AJ19, N02AJ17, N02AA55, N02AA56
2	Mirtazapine	716 (5.6)	N06AX11
3	Buprenorphine	642 (5.0)	N02AE01
4	Escitalopram	489 (3.8)	N06AB10
5	Risperidone	472 (3.7)	N05AX08
6	Sertraline	432 (3.4)	N06AB06
7	Temazepam	378 (2.9)	N05CD07
8	Morphine	376 (2.9)	N02AA01, N02AG01, N02AA51, N02AA04
9	Codeine	375 (2.9)	N02AJ01, N02AJ02, N02AJ03, N02AJ06, N02AJ07, N02AJ08, N02AJ09, N02AA55, N02AA56, N02AA59
10	Citalopram	329 (2.6)	N06AB04
11	Quetiapine	326 (2.5)	N05AH04
12	Pregabalin	319 (2.5)	N03AX16
13	Oxazepam	312 (2.4)	N05BA04
14	Melatonin	299 (2.3)	N05CH, N05CH01
15	Amitriptyline	246 (1.9)	N06AA09, N0CA01
16	Venlafaxine	206 (1.6)	N06AX16
17	Valproate	205 (1.6)	N03AG01
18	Diazepam	186 (1.4)	N05BA01
19	Olanzapine	180 (1.4)	N05AH03
20	Prochlorperazine	158 (1.2)	N05AB04

## Data Availability

The MedicineInsight data can be obtained from the Australian Commission on Safety and Quality in Health Care (QUMProgram@safetyandquality.gov.au).

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
