# Peer review of "Ten-Year Trends in Psychotropic Prescribing and Polypharmacy in Australian General Practice Patients with and without Dementia"

_jcm, 2023, doi:10.3390/jcm12103389_

Round 1
Reviewer 1 Report
Dear editor,
In the article entitled “Ten-year Trends in Psychotropic Prescribing and Polypharmacy in Australian General Practice Patients with and without Dementia”, authors describe the decreasing tendency on the prescription of antipsychotics and other psychotropic drugs in Australian patients with dementia.
Even though the topic is not original (see Kirkham et al., 2016 for a review), it is interesting to consider the use of these drugs in different countries and contexts. Nevertheless, I consider that the paper should be improved:
INTRODUCTION
Authors should reformulate the introduction, citing studies that have been published studying the same topic (vide supra). Also, they should talk about global numbers on dementia prevalence, not restricted to a sole country.
Furthermore, they should delve into the different therapeutic options for the treatment of dementia, not just naming the therapeutic groups.
METHODS
Why the authors did not consider epilepsy and insomnia among the matching covariates due to the fact that antiepileptic and hypnotic drugs are employed in dementia patients? (lines 97-103)
In table 1 (page 7), it would be helpful if Australian States are identified not only by the code, but by the complete name.
Also, between the “ACT” row and the “heart failure” row of the table 1 (page 7), there should be a more clear separation.
In table 1 (page 8), in the cell of the row “osteoporosis” and column “2011”-“control”, there may be a misprint.
Why the cells of the row “cancer” and columns “2014”-“control” and “dementia” are blank?
DISCUSSION
Authors formulate the discussion comparing their results with previous studies in Australia. They should reformulate the discussion to compare their results with the results of previous studies from other parts of the world due to the high ambition of their work.
Authors should delve into the fact that opioid prescription is increasing in people with dementia. There is any reason for that?
BIBLIOGRAPHY
There is a high number of references that are work documents or web pages. Authors should employ ideally pair-reviewed books or papers.
Kirkham J, Sherman C, Velkers C, Maxwell C, Gill S, Rochon P, Seitz D. Antipsychotic Use in Dementia. Can J Psychiatry. 2017;62(3):170-181. doi: 10.1177/0706743716673321.
Author Response
Manuscript ID: jcm-2352964
Title: Ten-year Trends in Psychotropic Prescribing and Polypharmacy in Australian General Practice Patients with and without Dementia
Journal: Journal of Clinical Medicine (JCM)
Section: Mental Health
Special Issue: Drug Treatment in Psychiatry: Progress and Challenges
Guest Editor
Dear Dr Jeffrey Fessel,
We are thankful for the comments and the opportunity to revise and re-submit our paper. We have carefully reviewed the comments and have revised the manuscript accordingly. Following this letter are the point-by-point responses highlighted in yellow. The changes made to the manuscript are shown in a separate track change document and uploaded to the system with the revised version of the manuscript. We hope the revised version is now suitable for publication and look forward to hearing from you soon.
With kind regards,
Woldesellassie M Bezabhe (PhD, MSc, BPharm)
Lecturer in Medication Safety, School of Pharmacy and Pharmacology, University of Tasmania
Reviwer 1: Comments and Suggestions for Authors
Dear editor,
In the article entitled “Ten-year Trends in Psychotropic Prescribing and Polypharmacy in Australian General Practice Patients with and without Dementia”, authors describe the decreasing tendency on the prescription of antipsychotics and other psychotropic drugs in Australian patients with dementia.
Even though the topic is not original (see Kirkham et al., 2016 for a review), it is interesting to consider the use of these drugs in different countries and contexts. Nevertheless, I consider that the paper should be improved:
INTRODUCTION
Point 1: Authors should reformulate the introduction, citing studies that have been published studying the same topic (vide supra). Also, they should talk about global numbers on dementia prevalence, not restricted to a sole country.
Furthermore, they should delve into the different therapeutic options for the treatment of dementia, not just naming the therapeutic groups.
Response 1: We highlighted the global dementia figure (see the first sentence in the Introduction section). The Introduction briefly covers the use of psychotropics in patients with dementia, the limitations of previous Australian data (reference 8-14), and the rationale for this study. The Kirkham et al. 2017 study is relatively old and restricted to only antipsychotics. One unique aspect of our work is that it covers a number of groups of CNS-active drugs (termed psychotropics) and not only antipsychotics, and their use in combination.
METHODS
Point 2: Why the authors did not consider epilepsy and insomnia among the matching covariates due to the fact that antiepileptic and hypnotic drugs are employed in dementia patients? (lines 97-103)
Response 2: Thanks for noting this; we have now listed epilepsy in the lines mentioned, and all variables used for matching, including epilepsy, are listed in Table 1. Insomnia was not flagged in the MedicineInsight dataset; therefore, it was not included as a matching variable. However, anxiety, treated with a similar class of medication, was included as a matching variable.
Point 3: In table 1 (page 7), it would be helpful if Australian States are identified not only by the code, but by the complete name.
Also, between the “ACT” row and the “heart failure” row of the table 1 (page 7), there should be a more clear separation.
Response 3: Yes, thank you. The Australian states and territories in Table 1 are now written in full text, and a subheading row “Comorbidities” is added between ATC and heart failure.
Point 4: In table 1 (page 8), in the cell of the row “osteoporosis” and column “2011”-“control”, there may be a misprint. Why the cells of the row “cancer” and columns “2014”-“control” and “dementia” are blank?
Response 4: Thank you. Sorry for this; the blank cells in the “osteoporosis” and “cancer” rows are now complete.
DISCUSSION
Point 5: Authors formulate the discussion comparing their results with previous studies in Australia. They should reformulate the discussion to compare their results with the results of previous studies from other parts of the world due to the high ambition of their work.
Authors should delve into the fact that opioid prescription is increasing in people with dementia. There is any reason for that?
Response 5: With due respect, we compared our findings with those of international studies (references 17, 28, 34, 35, 36, 37, and 42), which were cited in the Discussion section. Recognising and managing pain in patients with dementia has received substantial attention over the last ten years, after being under-recognised and under-managed. This could be the second reason for the increased opioid use.
BIBLIOGRAPHY
Point 6: There is a high number of references that are work documents or web pages. Authors should employ ideally pair-reviewed books or papers.
Response 6: We agree that peer-reviewed journal articles or books are preferred sources to cite. However, the government documents, guidelines and data dictionaries we referenced are the only sources and have not been published as peer-reviewed articles elsewhere.
Kirkham J, Sherman C, Velkers C, Maxwell C, Gill S, Rochon P, Seitz D. Antipsychotic Use in Dementia. Can J Psychiatry. 2017;62(3):170-181. doi: 10.1177/0706743716673321.

Reviewer 2 Report
In this article, "Ten-year trends in psychotropic prescribing...", the authors describe a population-based study of prescribing patterns among dementia patients and controls in an Australian general practice.
The hypothesis is clearly stated in the Introduction. The Methods are outlined in sufficient detail. The Results are described well in the figures and tables. The conclusions in the Discussion follow from the data. My suggestion is to include additional references that discuss other drugs that are helpful to manage neuropsychiatric symptoms:
J Cummings et al, "Role of donepezil in the management of neuropsychiatric symptoms of Alzheimer's disease and dementia with Lewy bodies," CNS Neuroscience & Therapeutics 2016
M Canevelli et al, "Sundowning in dementia: clinical relevance..." Frontiers in Med 2016
- In the Introduction, I recommended that the authors mention that Acetylcholinesterase Inhibitors are useful for managing neuropsychiatric symptoms of both AD and DLB (see Cummings et al, CNS Neurosci & Ther, 2016, “Role of donepezil in the management of neuropsychiatric.…” as one example). If this drug class was not included in this study, then their omission should be listed in the Discussion as a limitation of the study.
- In the Discussion, it would be important for the authors to explain that pain treatments (with opiates, acetaminophen, etc) are often indicated for preventing “sundowning” behaviors in older patients with dementia (see Canevelli et al, Front Med, 2016, “Sundowning in dementia”).
- The article by Canevelli also points out that melatonin is useful for prevention of sundowning. Many of us consider “polypharmacy” a good thing (ie use of analgesics, melatonin, and antidepressants) as a reasonable alternative to use of anti-psychotic agents in dementia patients who have neuropsychiatric symptoms. Depression is a common in the early stages of AD, VaD and DLB, so it is important not to under-treat it (see Boehlen et al, Intl J Geriatr Psychiatry, 2019, “Evidence for underuse and overuse of antidepressants in older adults: a large population-based study.” In this study, 77.3% of patients with depression were under-treated, compared to 41.7% who were over-treated.).
Author Response
Manuscript ID: jcm-2352964
Title: Ten-year Trends in Psychotropic Prescribing and Polypharmacy in Australian General Practice Patients with and without Dementia
Journal: Journal of Clinical Medicine (JCM)
Section: Mental Health
Special Issue: Drug Treatment in Psychiatry: Progress and Challenges
Guest Editor
Dear Dr Jeffrey Fessel,
We are thankful for the comments and the opportunity to revise and re-submit our paper. We have carefully reviewed the comments and have revised the manuscript accordingly. Following this letter are the point-by-point responses highlighted in yellow. The changes made to the manuscript are shown in a separate track change document and uploaded to the system with the revised version of the manuscript. We hope the revised version is now suitable for publication and look forward to hearing from you soon.
With kind regards,
Woldesellassie M Bezabhe (PhD, MSc, BPharm)
Lecturer in Medication Safety, School of Pharmacy and Pharmacology, University of Tasmania
Reviewer 2: Comments and Suggestions for Authors
In this article, "Ten-year trends in psychotropic prescribing...", the authors describe a population-based study of prescribing patterns among dementia patients and controls in an Australian general practice.
The hypothesis is clearly stated in the Introduction. The Methods are outlined in sufficient detail. The Results are described well in the figures and tables. The conclusions in the Discussion follow from the data. My suggestion is to include additional references that discuss other drugs that are helpful to manage neuropsychiatric symptoms:
J Cummings et al, "Role of donepezil in the management of neuropsychiatric symptoms of Alzheimer's disease and dementia with Lewy bodies," CNS Neuroscience & Therapeutics 2016
M Canevelli et al, "Sundowning in dementia: clinical relevance..." Frontiers in Med 2016
Point 7: In the Introduction, I recommended that the authors mention that Acetylcholinesterase Inhibitors are useful for managing neuropsychiatric symptoms of both AD and DLB (see Cummings et al, CNS Neurosci & Ther, 2016, “Role of donepezil in the management of neuropsychiatric.…” as one example). If this drug class was not included in this study, then their omission should be listed in the Discussion as a limitation of the study.
Response 7: It should be noted that the purpose of the study was not to examine acetylcholinesterase inhibitors use in patients with dementia. The focus was on the potentially inappropriate use of CNS-active drugs in these individuals. The use of acetylcholinesterase inhibitors in managing BPSD is now mentioned in the Introduction section “Patients with dementia with Lewy bodies may experience severe adverse reactions when taking antipsychotics. Acetylcholinesterase inhibitors (rivastigmine or donepezil) are the drug of choice for managing agitation, aggression or psychosis in these patients”. Not examining the use of acetylcholinesterase inhibitors is highlighted in the Discussion section “Acetylcholinesterase inhibitors can be prescribed to manage BPSD, and this study did not examine their use.”
Point 8: In the Discussion, it would be important for the authors to explain that pain treatments (with opiates, acetaminophen, etc) are often indicated for preventing “sundowning” behaviors in older patients with dementia (see Canevelli et al, Front Med, 2016, “Sundowning in dementia”).
Response 8: The article “Sundowning in dementia…” is interesting and clearly describes the worsening of BPSD during the late afternoon or early evening. The reasons proposed for the increased use of opioids are prescribing for managing BPSD (instead of antipsychotics) or pain management. See “Response 5” above.
Point 9: The article by Canevelli also points out that melatonin is useful for prevention of sundowning. Many of us consider “polypharmacy” a good thing (ie use of analgesics, melatonin, and antidepressants) as a reasonable alternative to use of antipsychotic agents in dementia patients who have neuropsychiatric symptoms. Depression is a common in the early stages of AD, VaD and DLB, so it is important not to under-treat it (see Boehlen et al, Intl J Geriatr Psychiatry, 2019, “Evidence for underuse and overuse of antidepressants in older adults: a large population-based study.” In this study, 77.3% of patients with depression were under-treated, compared to 41.7% who were over-treated.).
Response 9: Thank you. We included this in the Discussion section “We acknowledge psychotropic polypharmacy is not always inappropriate in patients with dementia, given that conditions, including depression and pain, have often been under-managed in these patients.”

Round 2
Reviewer 1 Report
Authors have answered to all my questions and comments. Thank you so much.
Author Response
Manuscript ID: jcm-2352964
Title: Ten-year Trends in Psychotropic Prescribing and Polypharmacy in Australian General Practice Patients with and without Dementia
Journal: Journal of Clinical Medicine (JCM)
Section: Mental Health
Special Issue: Drug Treatment in Psychiatry: Progress and Challenges
Guest Editor
Dear Dr Jeffrey Fessel,
We are thankful for the comments and the opportunity to revise and re-submit our paper. We have carefully reviewed the comments and have revised the manuscript accordingly. Following this letter are the point-by-point responses highlighted in yellow. The changes made to the manuscript are shown in a separate track change document and uploaded to the system with the revised version. We hope the revised version is now suitable for publication and look forward to hearing from you soon.
With kind regards,
Woldesellassie M Bezabhe (PhD, MSc, BPharm)
Lecturer in Medication Safety, School of Pharmacy and Pharmacology, University of Tasmania
The study and results reported in this Australian study are valuable and will make a really worthwhile contribution to the existing literature. I have some suggestions that will improve the article by making it more clear that this topic has been widely studied internationally but less so in Australia. Further, there are two strengths that certainly need emphasis.
Point 1: I suggest starting the Discussion with a short paragraph along the following lines.
“Although not extensively examined in Australia, the problem of psychotropic prescribing for patients with dementia has been extensively reported in international studies (refs 17, 38, 42). The issue of treatment-resistance in major depression was examined in 65 randomized controlled trials performed in many countries, involving 12,415 patients (Nunez n, et al. J Affect Disord. 2022;302:385-400). Further, van der Spek et al developed an index that measured the appropriateness of psychotropic drug use in patients with dementia (van der Spek et al. J Clin Epidemiol.2015;68:9030912) that was implemented in a study of 380 patients for whom intervention, based upon use of that index, was shown to produce favorable results ((van der Spek et al. Age Ageing. 2018;47:430-437)”.
Response 1: Thanks for suggesting this; we have included the suggested paragraph at the start of the Discussion.
Point 2: Add (“....”) to Discussion para 4 line 3: an international meta-analysis “involving 25 studies from 12 countries” defining psychotropic polypharmacy....etc.”
Add (“....”) to Discussion para 5 line 7: a recent “cross-national European” study...etc.
Response 2: Thanks for recommending this. We have added the proposed texts.
Point 3: Add (“....”) to the last paragraph of the Discussion: “It is a notable strength of the presented study, that the controls and patients were matched by using a most extensive list of co-morbidities; and a particular strength is the finding that in those age ≥85, those with dementia had a ~25% rate of psychotropic polypharmacy as compared with ~12.5% in the controls (fig 4B”.
Response 3: We have added the above texts at the end of the Discussion.
“It is a notable strength of the present study that the controls and patients were matched by using a most extensive list of co-morbidities; and a particular strength is the finding that in those age ≥85, those with dementia had a ~25% rate of psychotropic polypharmacy as compared with ~15% in the controls (Figure 4B).”
